# Translation, Cultural Adaptation, and Validation of the Spanish Version of the Person-Centred Practice Inventory-Staff (PCPI-S)

**DOI:** 10.3390/healthcare12232485

**Published:** 2024-12-09

**Authors:** Ana Carvajal-Valcárcel, Edgar Benitez, Marta Lizarbe-Chocarro, María José Galán-Espinilla, Mónica Vázquez-Calatayud, Begoña Errasti-Ibarrondo, Ana Choperena, Brendan McCormack, Vaibhav Tyagi, Virginia La Rosa-Salas

**Affiliations:** 1Facultad de Enfermería, Universidad de Navarra, 31008 Pamplona, Spain; acarvajal@unav.es (A.C.-V.); mlizarbe@unav.es (M.L.-C.); achoperena@unav.es (A.C.); vlarsal@unav.es (V.L.R.-S.); 2Instituto de Investigación Sanitaria de Navarra (IdiSNA), 31008 Pamplona, Spain; mvazca@unav.es; 3Instituto de Ciencia de los Datos e Inteligencia Artificial (DATAI), Universidad de Navarra, 31009 Pamplona, Spain; ebenitezs@unav.es; 4Tecnun Escuela de Ingeniería, Universidad de Navarra, 20018 San Sebastián, Spain; 5Centro de Salud de Ultzama, Servicio Navarro de Salud-Osasunbidea, 31003 Pamplona, Spain; mj.galan.espinilla@navarra.es; 6Gerencia de Atención Primaria de Navarra, Servicio Navarro de Salud-Osasunbidea, 31003 Pamplona, Spain; 7Área de Desarrollo Profesional e Investigación en Enfermería, Clínica Universidad de Navarra, 31008 Pamplona, Spain; 8Susan Wakil School of Nursing and Midwifery, University of Sydney, Sydney 2050, Australia; brendan.mccormack@sydney.edu.au (B.M.); vaibhav.tyagi@sydney.edu.au (V.T.); 9Faculty of Medicine and Health, The University of Sydney, Sydney 2050, Australia

**Keywords:** cross-cultural adaptation, healthcare provider, measures, PCPI-S, person-centred care, person-centred practice, psychometric properties

## Abstract

**Background:** Person-centredness, a global movement in healthcare, is consistent with international developments in healthcare policy. It is important to have instruments to measure person-centred care. The Person-Centred Practice Inventory-Staff (PCPI-S) is an internationally recognized instrument that aims to measure how healthcare staff experience person-centred practice. **Aim:** To perform the cultural adaptation and psychometric testing of a Spanish version of the PCPI-S (PCPI-S (Sp)). **Method:** A two-stage research design was implemented as follows: (1) the translation and cultural adaptation of the PCPI-S from English to Spanish using the “Translation and Cultural Adaptation of Patient Reported Outcomes Measures-Principles of Good Practice”; (2) a quantitative cross-sectional survey for the psychometric evaluation of the PCPI-S. Test–retest reliability was evaluated using the Kendall tau concordance coefficient, internal reliability was assessed through the ordinal theta (OT) coefficient, and confirmatory factor analysis was performed to examine the theoretical measurement model. **Results**: A Spanish version of the PCPI-S was obtained. There were no significant difficulties in the translation process or the consulting sessions. A sample of 287 healthcare professionals participated in the study at least once. All the items showed at least a fair level of test–retest reliability. The OT scores were adequate (>0.69). The model showed good to adequate levels of fit: CFI = 0.89, SRMR = 0.068; RMSEA = 0.060 with CI90% (0.056–0.063). **Conclusions:** The Spanish translation of the PCPI-S was psychometrically valid when tested with Spanish healthcare professionals. This instrument will help identify professionals’ perceptions of person-centred practice, track the evolution of this practice over time, and assess interventions aimed at improving person-centred practice.

## 1. Introduction

Person-centredness has become a global movement in healthcare [1,2] and is consistent with international developments in healthcare policy [3]. In Crossing the Quality Chasm: A New Health System Report, the Institute of Medicine (IOM) emphasized the centrality and importance of person-centredness as one of the six major aims for improvement [4]. More recently, the World Health Organization (WHO) in the Sixty-Ninth World Health Assembly [5] stated that developing more integrated people-centred care systems has the potential to generate significant benefits in healthcare, including “improved access to care, improved health and clinical outcomes, better health literacy and self-care, increased satisfaction with care, improved job satisfaction for health workers, improved efficiency of services, and reduced overall costs” [5] (p. 2). Furthermore, the juxtaposition of person-centred integrated care models with a health promotion/public health approach has recently been identified as a possible means to improve health and wellbeing outcomes for individuals, communities, and populations [6,7].

It is widely acknowledged that person (and family)-centred care focuses on the whole person as a unique individual and not just on his or her illness or disease. In viewing the individual through this lens, healthcare providers come to know and understand the person’s life story, his or her experience of health, the role of the family in the person’s life, and the role they may play in supporting the person to achieve health outcomes [8]. Likewise, person-centred health systems enable individuals to make informed decisions about and to successfully manage their own health and care [9]. In addition, evidence shows that meeting the needs of the patient within a person-centred care (PCC) approach is imperative for an efficient and effective healthcare system [10,11]. This requires healthcare services to work in partnership to deliver care responsive to people’s individual abilities, preferences, lifestyles, and goals [9].

As person-centredness is central to healthcare, it has often been conceptualized as a singular concept in different frameworks and models. These include relationship-centred care [12], compassionate care [13], and dignified care [14]. However, none of these models adopt a holistic perspective of person-centredness or integrate the main attributes that help to understand the person as a whole. In this regard, the person-centred practice framework (PCPF) developed by McCormack and McCance [3] is an internationally recognized theoretical framework that assists healthcare professionals in fully understanding all the dimensions that integrate person-centredness and how these dimensions can be operationalized in practice [15]. This framework has been applied for more than a decade, and although it was designed in the field of nursing practice, it is currently situated within healthcare systems more broadly [16].

The PCPF includes four main domains: *prerequisites*, *the care environment*, *person-centred processes,* and *person-centred outcomes* (Figure 1 [3]). The four domains are nested within a fifth domain, *the macro context* of the person-centred health setting [16]. As shown in Table 1, the domains include their corresponding defining constructs. From this point of view, staff, who are an essential element of the *prerequisites* domain, are considered necessary to manage the *care environment* to provide effective care by means of *person-centred processes*. This process leads to the achievement of the outcome of a *healthful culture*, which is the central domain of the framework. Furthermore, this framework goes a step further by suggesting that a healthful caring culture should focus not only on the person receiving care but also on staff well-being. As such, the framework also asserts that the environment in which healthcare is experienced should ensure the individual health and well-being of all those involved in PCC [16].

Based on the PCPF, three instruments have been developed: The Person-Centred Practice Inventory-Staff (PCPI-S) [15], the Person-Centred Practice Inventory-Student (PCPI-ST) [17], and the Person-Centred Practice Inventory-Care (PCPI-C) [18].

Specifically, the PCPI-S was developed for healthcare staff with the aim of measuring how they experience person-centred practice. Paying attention to staff experiences is not common in other instruments that assess PCC [9,19]. As stated above, an additional strength of the PCPI-S is that unlike other instruments, it maps onto a theoretical framework [15,20]. Similarly, in contrast to other Spanish-translated instruments that are primarily used in elderly care services [19,21,22], the PCPI-S is designed to assess person-centred practice in a broader range of healthcare settings (hospitals, primary care, nursing homes, long-term care, etc.). Finally, this instrument has been validated in Norwegian [1], German [23], Malaysian [24], and Korean [25], showing good psychometric properties. However, it has not yet been translated and validated in Spanish, which is the fourth most spoken language in the world and the official language of most Latin American countries [26].

In sum, this tool is based on the PCPF, measures the key elements of the culture and context that contribute to the development of person-centred practice from the perspective of healthcare professionals, and can be used in different healthcare settings [3]. Accordingly, this instrument is anticipated to enable practitioners to evaluate their level of person-centred practice, thereby serving as a foundation for the development and implementation of targeted interventions aimed at advancing their professional growth. Likewise, this instrument will assist managers and supervisors across diverse healthcare settings by enabling the assessment of PCC levels among healthcare professionals, while also identifying barriers and facilitators to the delivery of PCC. This comprehensive understanding will support the design, implementation, and evaluation of interventions aimed at enhancing person-centred practices at individual, environmental, and care process levels. Hence, there is significant interest in developing a translated and validated Spanish version of the PCPI-S. This would facilitate the conduct of interterritorial studies within Spain, a country characterized by the coexistence of multiple regional languages, while Spanish serves as the common official language. This interest stems from the growing focus on developing PCC policies, programs, approaches, and/or initiatives in Spain, all aimed at the humanization of care [27,28,29,30,31,32,33].

## 2. Methods

The aim of this study was to carry out the cultural adaptation and psychometric testing of a Spanish version of the PCPI-S (PCPI-S (Sp)).

The following two-stage research design was used: (1) translation and cultural adaptation of the PCPI-S from English to Spanish and (2) a quantitative cross-sectional survey for the psychometric evaluation of the PCPI-S.

Description of the Instrument

The PCPI-S consists of 59 items drawn from 17 constructs of the PCPF [15]. All the items are measured on a five-point Likert scale (1-strongly disagree to 5-strongly agree). The PCPI-S has shown good psychometric properties in the following different languages: Norwegian [1] (α = 0.88–0.93), German [23] (α = 0.90–0.94), Malaysian [24] (α = 0.59–0.86), and Korean [25] (α = 0.95). The aim of the instrument is to measure healthcare professionals’ perceptions of PCC in any healthcare context. This instrument provides a numerical score, but it does not aim to quantify how healthcare professionals provide PCC. It guides how professionals perceive PCC [15].

### 2.1. Phase 1: Translation and Cultural Adaptation

Participants

The participants in this phase included the principal investigator, the author of the questionnaire, two forward translators (Spanish native speakers fluent in English from the healthcare context and with knowledge of PCC), two back translators (English native speakers fluent in Spanish), an expert in humanities to correct lexicon-semantic errors, an expert in psychometric properties, a panel of six experts, and a group of five healthcare professionals who tested the instrument.

Description of the Procedure

The PCPI-S was translated into Spanish following the guide of the “Translation and Cultural Adaptation of Patient Reported Outcomes Measures-Principles of Good Practice” [34]. The steps that were carried out according to this guide were as follows:Preparation: The principal investigator contacted one of the authors of the questionnaire.Forward translation: The forward translators translated the questionnaire from English to Spanish (Spanish version or SV) (SV_1_ and SV_2_).Reconciliation: From the two forward translations (SV_1_ and SV_2_) of the questionnaire, a single forward translation in Spanish (SV_3_) was obtained by consensus of a committee composed of the forward translators, the principal investigator, and an expert in PCC.Back translation: The back translators translated the new Spanish version (SV_3_) into English (English version or EV) (EV_1_ and EV_2_).Back translation review: The committee that participated in the reconciliation obtained a new English version (EV_3_) of the questionnaire.Harmonization: The EV_3_ was compared with the English translation of the Norwegian versions of the PCPI-S.Cognitive debriefing: An expert group—heterogeneous in terms of academic education, academic discipline, and person-centred practice education—assessed the clarity and relevance of the SV_3_ using a Likert scale from 1 (not clear or relevant) to 4 (strongly clear or relevant). Likewise, a small group of healthcare professionals tested the instrument. Both the expert group and the healthcare professionals were asked about their perception after completing the questionnaire. The time for completion of the questionnaire was also registered.Review of the cognitive debriefing results: To assess the content validity of the instrument, the content validity index (I-CVI) and the scale content validity (S-CVI/Ave) were calculated. A content analysis of the comments suggested by the experts and the healthcare professionals was carried out. The author of the questionnaire was asked to clarify discrepancies.Proofreading: One of the study researchers who is an expert on humanities corrected the lexicon-semantic (words), morphosyntactic (concordance among gender and verb tense), and orthographic errors from the last version achieved in consensus with the project manager.Final report: The principal investigator wrote the final report, including the process and methodologies used.
Data Analysis

Clarity and relevance were determined by the item content validity index (I-CVI), which assesses the quality and relevance of each individual item, and the scale content validity index (S-CVI/Ave), which refers to the extent to which an instrument adequately captures the domain or construct it is intended to measure. To obtain the I-CVI, the number of observers who scored the items with 3 or 4 was divided by the total number of observers. Moreover, the kappa statistic was calculated to identify the degree of agreement beyond chance. The results were interpreted according to the Cicchetti and Sparrow classification [35]. The content validity of the entire instrument (S-CVI/Ave) was calculated as the average of the I-CVIs and was interpreted as excellent if the value obtained was greater than 0.90 [36].

### 2.2. Phase 2: Psychometric Properties

Settings and Participants

The instrument was tested in a sample of health professionals from seven different settings: four hospitals (two private *Clínica Universidad de Navarra* and *Clínica San Miguel*) and two public (*Hospital Universitario de Navarra* and *Hospital Reina Sofía*)), primary care, and two public nursing homes (*Amma Mutilva* and *Amma Oblatas*).

The sample comprised all healthcare professionals who were working within those settings with patients and families, who had any experience with PCC and who were willing to participate in the study.

Ethical Committee

The study was approved by the Ethics Committee on Clinical Research (CEIC) at Universidad de Navarra (2021.125 Mod1) and the Ethics Committee on Clinical Research at Departamento de Salud, Gobierno de Navarra (PI_2021/141). One of the authors of the PCPI-S (B.M.), who is part of the project, provided authorization to validate the instrument.

Description of the Procedure

Once the manager of the centers provided the authorization, the questionnaire was disseminated online via the weekly newsletter of the hospitals, leaflets in the units, or the unit managers. The gatekeeper of each setting promoted participation in the study. All healthcare professionals willing to participate had access to the information sheet, questionnaire, and consent form online. The participants who agreed to participate provided the following information: Age, sex, profession, years of experience, and setting. Then, they completed the questionnaire. Those participants who wanted to complete the questionnaire for the retest provided their email address as well as their consent to contact them. Three weeks later, an email was sent to them with the questionnaire link and one week later, a reminder.

Sample Size

Based on the guidelines provided by MacCallum, Browne, and Sugawara (1996) [37], the required sample size to achieve a statistical power of 80% was calculated.

The analysis used the following input parameters: α = 0.05 (level of significance), RMSEA (null hypothesis) = 0.05 (close fit under the null hypothesis), RMSEA (alternative hypothesis) = 0.08 (adequate fit under the alternative hypothesis), and df = 1364 (degrees of freedom). The results indicated that a minimum sample size of 200 was necessary to meet the desired level of statistical power. This analysis confirms that the sample size used in the study is adequate for the analyses conducted.

Statistical Analysis

Preliminary, skewness, and kurtosis statistics were evaluated for all items (Appendix A). Sixteen items were identified to have absolute values above the two units considered as the cut-off point for an approximation to a normal distribution [38]. Starting from this deviation from the normal approximation, the test–retest validity was evaluated using the Kendall tau concordance coefficient [39,40]. In the same sense, the correlation coefficients shown in Appendix A are polychoric. The internal reliability evaluation of the constructs was performed using the ordinal theta (OT) coefficient, proposed by Zumbo et al. [41].

In accordance with the two-step approach proposed by Anderson and Gerbing for performing factor analysis and evaluating causal relationships [42], confirmatory factor analysis (CFA) was performed using the simplest possible model, where all factors were correlated. The estimation of the CFA was performed using the robust two-stage maximum likelihood (RML) method [38]. For the validation and evaluation of the fit, the values proposed by Hu and Bentler (1999) [43] were used: 1) comparative fit index (CFI) > 0.95 and standardized root mean square residual (SRMR) < 0.09 or 2) root mean square error of approximation (RMSEA) < 0.05 and SRMR < 0.06. Modifications to the model and the restriction or release of parameters were performed using Wald’s test and Lagrange multipliers, respectively. Changes were accepted if the variations in χ^2^ values were significant at *p* < 0.05, and if there was a theoretical justification for said modification.

Modifications to the model were made one at a time. The reliability and validity of the constructs were evaluated in the final model with the following goodness-of-fit criteria: item reliability >0.39, composite reliability >0.7, and variance extracted >0.49 units [44,45]. Convergent validity was assessed by *t*-tests on parameters and discriminant validity on the ±2 standard errors around the correlation between factors. Data analysis was performed using SAS version 9.4.

## 3. Results

### 3.1. Phase 1: Translation and Cultural Adaptation

Preparation: Permission for the translation of the questionnaire was obtained from the author of the questionnaire (BM).Forward translation: Two forward translations from English to Spanish were performed (Spanish version or SV) (SV_1_ and SV_2_) by two fluent speakers. The mean time for the translation was two hours each.Reconciliation: A new Spanish version (SV_3_) merged from the two forward translations (SV_1_ and SV_2_) was obtained. Consensus was achieved by a committee composed of the principal investigator and the forward translators. They resolved discrepancies between the forward translation (e.g., they clarified the term “evidence”; they discussed the Spanish word “feedback” that was maintained in English because it is also used in the Spanish context; and they considered it appropriate to use the word “care” in the whole questionnaire due to the similar meaning of the term “care” for all healthcare professionals).Back translation: Two back translations were performed (English version or EV) (EV_1_ and EV_2_). The translations were conceptual rather than literal to maintain the key concepts of the questionnaire. The back translators did not have problems carrying out the translation.Back translation review: The committee obtained a new English version of the questionnaire (EV_3_). This version was revised by the author of the questionnaire, and it was suggested that 12 items be modified.Harmonization: The author of the questionnaire did not identify significant differences among the versions.Cognitive debriefing: A group of six healthcare professionals participated in a panel of experts. They were clinical, academic–clinical, and health managers. Four of them had knowledge and experience with PCC. One of them had experience in psychometric validation. Moreover, five healthcare professionals (one physician, three nurses, and one assistant nurse) completed the questionnaire in an average time of six minutes.Regarding clarity, 53/59 of the items had an excellent I-CVI, and 6/59 had an acceptable I-CVI (Table 2). The S-CVI/Ave was excellent, at 0.90. Regarding relevance, the I-CVI was excellent for all the items, and the S-CVI/Ave was 0.96.Review of the cognitive debriefing results: The minor changes suggested by the experts were made to items 18, 45, 47, 52, and 54 (Table 3). The healthcare professionals considered that the questionnaire was easy to understand and included all the relevant aspects of their clinical practice.Proofreading: Lexicon-semantic, morphosyntactic, and orthographic errors were corrected in the following five items: 12, 35, 38, 44, and 46.Final report: The final version of the questionnaire was obtained.

### 3.2. Phase 2: Psychometric Properties

Characteristics of the Participants

Data were gathered between September and November 2021. A total of 287 health professionals completed the PCPI-S at least once. One participant’s responses were withdrawn because the descriptive data were not completed. Table 4 shows the characteristics of the sample. The questionnaire was completed three weeks later by 114 participants (39.7% retention rate).

Psychometric Aspects

In the evaluation of skewness and kurtosis, items with values that fell outside the criteria defined for normality indicated high kurtosis and negative skewness, especially in Domain 1 (Prerequisites) (see Appendix A), showing a high coincidence of responses towards high values of the scale. Regarding the polychoric correlations, the correlations of the items in general showed independent patterns (low correlations) between Domain 1 and Domain 2 (the practice environment) and to a lesser extent with Domain 3 (person-centred processes). In the same sense, Domain 3 presented a pattern of correlation less with Domain 2 than with Domain 1. Within each domain, the correlation coefficients within Domain 3 (person-centred processes) were, in general, high compared to those in Domains 1 and 2.

In the test–retest reliability analysis, all the items showed fair levels of congruence (0.2–0.4), with half of the items reaching moderate levels (0.4–0.6) and 5% of items reaching substantially high levels (0.6–0.8) (see Appendix A).

In the internal reliability analysis, the constructs showed adequate OT values (>0.69); however, construct C17 showed multicollinearity (OT >0.9) (Table 5). The skewness and kurtosis values of the constructs remained within the predetermined limits (<|2|).

The CFA for the original PCPI-S [15] yielded fit values below the established cut-off points (Table 6). Therefore, it was decided to evaluate the modification indices with the aim of allowing adjustment of the instrument. The descriptive results were also considered (Table 5); the constructs with the lowest level of reliability were construct C5 “Clarity of beliefs and values” and construct C10 “Potential for innovation and risk taking”. In addition, the Wald test did not identify any parameter to release, contrary to the Lagrange multiplier test, which identified several dependencies of construct C10 “Potential for innovation and risk taking” that were not identified in the original factorial model, PCPI-S [15]. Within these relationships, the most relevant, according to the Lagrange multiplier test, were related to the influence of constructs C6 “Skill-Mix”, C11 “The physical environment”, and C13 “Working with patients’ beliefs and values”; of items 32 and 36; or of the items of construct C10 (items 33 and 35). Since these relationships were not supported by the PCPF and since C10 had the minimum number of items (three), for identifiability processes [44], the complete construct was removed from the original model [15]. After this construct was removed, the defined criteria for the RMSEA and SRMR values were achieved (Table 6).

The instrument that was obtained, which had 56 items and 16 factors, was evaluated for its measurement properties (Table 7). Low variance extraction values were observed for constructs C1 “Professionally competent”, C2 “Developed interpersonal skills”, C3 “Being committed to the job”, C5 “Clarity of beliefs and values”, C6 “Skill mix”, and C7 “Shared decision-making systems”. For constructs C1 “Professionally competent”, C5 “Clarity of beliefs and values”, and C6 “Skill-mix”, the composite reliability values were also below the criteria (<0.7). The item reliabilities were also low for some of the items in these constructs. Regarding the analysis of convergent validity, it was only found that the correlation between C5 “Clarity of beliefs and values” and C6 “Skill-Mix” presented a value statistically equal to one, r = 0.94 with 95% CI (0.84–1.03).

## 4. Discussion

The first version of the PCPI-S culturally adapted and validated in Spanish (PCPI-S-Sp) was obtained. The PCPI-S has been widely used and validated in different countries [1,19,20,21]. Having the Spanish version of the instrument will reinforce and facilitate its dissemination in other contexts. Furthermore, this instrument will allow us to explore person-centred practice at the individual, unit, and organizational levels [15] in the Spanish context. It is important to note that the validation of the PCPI-S in Spain serves as the starting point for its applicability in the other 20 countries where Spanish is the official language. However, linguistic and cultural differences may impact its applicability, requiring further adaptations and validations in each of these contexts. Conducting multi-country validation studies or cross-cultural research in diverse healthcare settings would be essential to confirm the tool’s psychometric robustness and cultural adaptability across various contexts, thereby establishing its generalizability [46].

Unlike other instruments measuring PCC that are focused on singular aspects of person-centredness [12,13,14] or that are not underpinned by a framework [15,20], the PCPI-S is based on the PCPF [3]. It is important to highlight that this framework defines the main aspects of person-centredness and that it has recently been culturally adapted into Spanish [47].

Moreover, the PCPI-S was developed to be used by different healthcare professionals (physicians, nurses, assistant nurses, etc.) [15]. On the one hand, this helps identify the perceptions of different professionals regarding person-centred practice, and on the other hand, it helps generate evidence that providing a PCC needs an interdisciplinary perspective [48].

The Spanish version of the PCPI-S was the result of a complex process of translation and adaptation following the “*Translation and cultural adaptation process for Patient-Reported Outcomes Measures-Principles of Good Practice*” [34]. This guide is widely used and internationally recognized. It was also used to translate and adapt the PCPI-S into Norwegian [1] and German [23]. Following the same guide for the cultural adaptation of three different versions of the PCPI-S provides consistency among the different versions [49]. Moreover, it allows the comparison of the results of the different versions measuring the same phenomenon.

One strength of the study is that the author of the PCPI-S, who is also the author of the PCPF, took part in the process of translation and cultural adaptation. This allowed us to clarify questions about the instrument at any time in the process to maintain the conceptual meaning of the items rather than the literal meaning. The literature shows that this is one of the most important requisites to reach quality during the process of translation and cultural adaptation [34].

The term “care”, which is key in PCC, appears throughout the entire questionnaire. This term is mainly associated with the field of nursing. In fact, in the translation of the PCPI-S for the Norwegian context [1], it was found that this term is not applicable to other professions such as medicine or psychology. In the present study, the healthcare professionals from the panel of experts concluded that in the Spanish cultural context, this term is generalizable to all healthcare professionals. For this reason, as in the original version of the PCPI-S, the term “care” (*cuidado* in Spanish) was maintained in the Spanish version.

In the present study, no difficulties in the translation, cultural adaptation, and validation of the PCPI-S were encountered, consistent with other studies that have translated the instrument into other languages, such as Norwegian [1], German [23], Malaysian [24], and Korean [25]. The validation of the same instrument in different languages will allow us to compare international findings about how person-centred practice is provided in different countries. However, there are some issues that deserve consideration, as shown below.

From the psychometric evaluations, some items exhibited low retest reliability, ranging between 0.2 and 0.35. For example, in the ‘Prerequisites’ dimension, items 2, 8, and 11 demonstrated this issue, as did items 29 and 36 in the ‘Care environment’ dimension. Similarly, in the ‘Care Processes’ dimension, items 50 and 55 showed low reliability. This may be attributed to the nature of the aspects being assessed, which are related to the relationships with patients, colleagues, and other professionals. These factors are expected to evolve over time. Therefore, it is natural for the reliability of these elements to vary in repeated tests, as they are neither constant nor static [50].

On the other hand, construct C10 “Potential for innovation and risk taking” already showed problems by presenting low reliability indices and low correlations. This was reflected in their inability to evaluate this construct, and in this way, other constructs began to show influences on this construct, such as “skill mix” and “power sharing” (e.g., *Skill Mix* influencing answers related to whether *support is given to do different things that improve the practice*). This correlation is supported conceptually since it is apparent that when a skill mix is present, the potential for innovation and risk taking is higher [44]. The literature shows that one important element in healthcare is how a good ratio of skill mix has a positive influence on complex and innovative decisions according to the patient’s needs. When power sharing is elicited more effectively, the team involves patients and incorporates important information for and from patients in deliberations, and therefore, more balanced and appropriate decisions are made. As a consequence, interprofessional shared decision-making is associated with improved processes and outcomes and decreased risks. Some examples of this are the studies made by different authors [51,52,53]. In future works, different ways of evaluating this construct should be explored to identify possible correlations with the different constructs that have not been identified.

Previous research has shown that quality of care improves when the employees within a healthcare team differ in terms of education, expertise, and competencies. As shown in Aiken and colleagues’ review [54], a rich nurse skill mix (a higher percentage of professional nurses among all nursing personnel) was associated with better quality of care, higher safety, and higher hospital ratings from patients.

Additionally, in the psychometric testing, a strong correlation between Construct 5, “Clarity of Beliefs and Values”, and Construct 6, “Skill Mix”, was found, presenting a value statistically equal to one, r = 0.94. This correlation is expected within the theoretical framework underpinning this study, the PCPF by McCormack et al. (2021) [16]. According to this framework, clarity of beliefs and values, particularly the shared commitment to placing the patient at the center of care, is intrinsically linked to the composition and expertise of the care team (skill mix) [55]. This synergy ensures that the team functions cohesively to meet the patient’s needs effectively.

However, the magnitude of the correlation (r = 0.94) raises potential concerns about conceptual overlap. While the constructs are theoretically distinct, the high correlation suggests they may not be entirely independent. To address this, future iterations of the tool could refine the items to emphasize the structural aspects of “Skill Mix” and the relational dynamics within “Clarity of Beliefs and Values”. Such adjustments would preserve the constructs’ theoretical foundations while enhancing the tool’s ability to distinguish between them.

Despite this, the findings underscore the importance of integrating these two constructs in the delivery of PCC. As highlighted by the Dutch Health Care Inspectorate, ensuring that the quantity and expertise of staff are aligned with the healthcare needs of patients is critical for providing high-quality care [56,57]. The optimal fit between professionals and the skill mix—while focused on patient needs—is essential for delivering safe and effective care.

Another finding in this study was the low reliability values of the constructs C1 “Professionally competent” and C6 “Skill Mix”. International research shows that when professionals perceive themselves to be professionally competent, interprofessional collaboration is essential to obtain an adequate skill mix [58]. However, one barrier that could emerge is that professionals may lack understanding of the specific professional roles and responsibilities of the different professionals [58]. Therefore, they lack knowledge about the contribution of other professionals in the patient’s care and hence the impact of the whole team on the patient’s health [59].

Finally, although the recruitment was not probabilistic, participants from various work environments and with a wide age range were still included. This improved the external validity of the findings reported in this study. However, some caution must be taken when using this instrument. Because the instrument was developed for different healthcare professionals, it would be advisable to test the instrument with larger heterogeneous samples.

In general, the measurement instrument shows adequate adjustment values. It is to be expected that, being such a complex instrument, the adjustment indices show nonconformities; however, there were few steps followed to achieve the aforementioned adjustment, and the required level was easily reached.

## 5. Conclusions

In this research, the first Spanish version of the PCPI-S (Sp), which assesses the perception of healthcare professionals about a person-centred practice, was developed. Having this tool in Spanish will increase the interest in developing PCC policies, programs, interventions, approaches, and/or initiatives in Spain.

The PCPI-S has adequate psychometric properties for the measurement of person-centred practice in the Spanish context. This will facilitate the identification of different professionals’ perceptions of person-centred practice, tracking the evolution of this practice over time and assessing interventions that seek to improve person-centred practice. This version of the PCPI-S will allow the development of international comparative studies in different countries.

Furthermore, the existence of two additional instruments based on the same framework as the PCPI-S, namely, the PCPI-ST (students’ perceptions) and the PCPI-C (patients’ perceptions), will allow us to measure the same phenomenon from the perspective of all those involved in PCC, providing a global overview of the phenomenon. These instruments are currently being culturally adapted and validated in Spanish.

## Figures and Tables

**Figure 1 healthcare-12-02485-f001:**
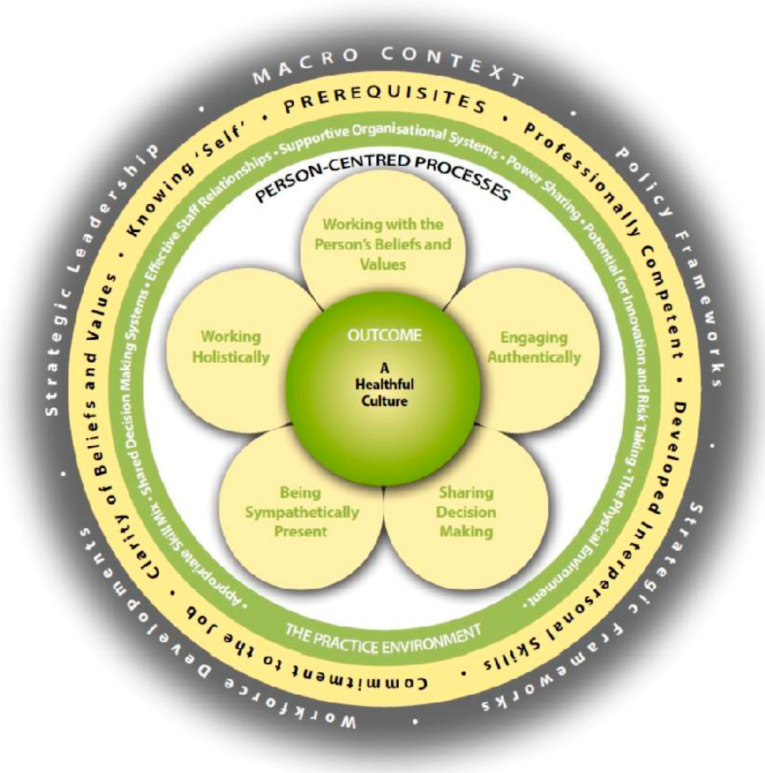
*Person Centred Practice Framework* (PCPF) by McCormack, B., McCance, T. (2021) [3,16].

**Table 1 healthcare-12-02485-t001:** Theoretical framework that underpins the PCPI-Staff and the items of each construct.

Dimensions	Constructs	Items
Prerequisites	1. Professionally competent	3
	2. Developed interpersonal skills	4
	3. Being committed to the job	5
	4. Knowing self	3
	5. Clarity of beliefs and values	3
Care environment	6. Skill mix	3
	7. Shared decision-making systems	4
	8. Effective staff relationships	3
	9. Power sharing	4
	10. Potential for innovation and risk taking	3
	11. The physical environment	3
	12. Supportive organizational systems	5
Care processes	13. Working with patients’ beliefs and values	4
	14. Shared decision-making	3
	15. Engagement	3
	16. Having sympathetic presence	3
	17. Providing holistic care	3
Total items		59

**Table 2 healthcare-12-02485-t002:** Clarity and relevance of the Spanish version of the Person-Centred Practice Inventory-Staff (PCPI-S).

		Clarity			Relevance		
		I-CVI	K *	Evaluation	I-CVI	K *	Evaluation
	**Professionally competent**						
Item 1	I have the necessary skills to negotiate care options	1	1		1	1	Excellent
Item 2	When I provide care I pay attention to more than the immediate physical task	1	1	Excellent	1	1	Excellent
Item 3	I actively seek opportunities to extend my professional competence	0.83	0.81	Excellent	0.83	0.81	Excellent
	**Developed interpersonal skills**						
Item 4	I ensure I hear and acknowledge others’ perspectives	0.67	0.57	Acceptable	1	1	Excellent
Item 5	In my communication I demonstrate respect for others	0.83	0.81	Excellent	1	1	Excellent
Item 6	I use different communication techniques to find mutually agreed solutions	0.83	0.81	Excellent	0.83	0.81	Excellent
Item 7	I pay attention to how my non-verbal cues impact on my engagement with others	1	1	Excellent	1	1	Excellent
	**Being committed to the job**						
Item 8	I strive to deliver high-quality care to people	0.83	0.81	Excellent	0.67	0.57	Acceptable
Item 9	I seek opportunities to get to know the person and their family in order to provide holistic care	0.83	0.81	Excellent	1	1	Excellent
Item 10	I go out of my way to spend time with people receiving care	1	1	Excellent	1	1	Excellent
Item 11	I strive to deliver high-quality care that is informed by evidence	0.83	0.81	Excellent	0.83	0.81	Excellent
Item 12	I continuously look for opportunities to improve the care experiences	0.67	0.57	Acceptable	0.83	0.81	Excellent
	**Knowing self**						
Item 13	I take time to explore why I react as I do in certain situations	1	1	Excellent	1	1	Excellent
Item 14	I use reflection to check out if my actions are consistent with my ways of being	0.83	0.81	Excellent	0.83	0.81	Excellent
Item 15	I pay attention to how my life experiences influence my practice	1	1	Excellent	1	1	Excellent
	**Clarity of beliefs and values**						
Item 16	I actively seek feedback from others about my practice	0.83	0.81	Excellent	1	1	Excellent
Item 17	I challenge colleagues when their practice is inconsistent with our team’s shared values and beliefs	1	1	Excellent	1	1	Excellent
Item 18	I support colleagues to develop their practice to reflect the team’s shared values and beliefs	0.83	0.81	Excellent	1	1	Excellent
	**Skill mix**						
Item 19	I recognize when there is a deficit in knowledge and skills in the team and its impact on care delivery	1	1	Excellent	1	1	Excellent
Item 20	I am able to make the case when the skill mix falls below acceptable levels	0.83	0.81	Excellent	0.83	0.81	Excellent
Item 21	I value the input from all team members and their contributions to care	1	1	Excellent	1	1	Excellent
	**Shared decision-making systems**						
Item 22	I actively participate in team meetings to inform my decision-making	0.83	0.81	Excellent	1	1	Excellent
Item 23	I participate in organization-wide decision-making forums that impact on practice	1	1	Excellent	1	1	Excellent
Item 24	I am able to access opportunities to actively participate in influencing decisions in my directorate/division	1	1	Excellent	1	1	Excellent
Item 25	My opinion is sought in clinical decision-making forums (e.g, ward rounds, case conferences, discharge planning)	1	1	Excellent	1	1	Excellent
	**Effective staff relationships**						
Item 26	I work in a team that values my contribution to person-centred care	1	1	Excellent	1	1	Excellent
Item 27	I work in a team that encourages everyone’s contribution to person-centred care	1	1	Excellent	1	1	Excellent
Item 28	My colleagues positively role model the development of effective relationships	0.83	0.81	Excellent	0.83	0.81	Excellent
	**Power sharing**						
Item 29	The contribution of colleagues is recognized and acknowledged	0.67	0.57	Acceptable	1	1	Excellent
Item 30	I actively contribute to the development of shared goals	1	1	Excellent	1	1	Excellent
Item 31	The leader facilitates participation	1	1	Excellent	1	1	Excellent
Item 32	I am encouraged and supported to lead developments in practice	1	1	Excellent	1	1	Excellent
	**Potential for innovation and risk taking**						
Item 33	I am supported to do things differently to improve my practice	1	1	Excellent	0.83	0.81	Excellent
Item 34	I am able to balance the use of evidence with taking risks	0.67	0.57	Acceptable	0.83	0.81	Excellent
Item 35	I am committed to enhancing care by challenging practice	0.83	0.81	Excellent	1	1	Excellent
	**The physical environment**						
Item 36	I pay attention to the impact of the physical environment on people’s dignity	1	1	Excellent	1	1	Excellent
Item 37	I challenge others to consider how different elements of the physical environment impact on person-centredness (e.g., noise, light, heat, etc.)	1	1	Excellent	1	1	Excellent
Item 38	I seek out creative ways of improving the physical environment	1	1	Excellent	0.83	0.81	Excellent
	**Supportive organizational systems**						
Item 39	In my team we take time to celebrate our achievements	1	1	Excellent	1	1	Excellent
Item 40	My organization recognizes and rewards success	1	1	Excellent	1	1	Excellent
Item 41	I am recognized for the contribution that I make to people having a good experience of care	1	1	Excellent	0.83	0.81	Excellent
Item 42	I am supported to express concerns about an aspect of care	1	1	Excellent	0.83	0.81	Excellent
Item 43	I have the opportunity to discuss my practice and professional development on a regular basis	1	1	Excellent	1	1	Excellent
	**Working with patients’ beliefs and values**						
Item 44	I integrate my knowledge of the person into care delivery	0.83	0.81	Excellent	1	1	Excellent
Item 45	I work with the person within the context of their family and carers	1	1	Excellent	1	1	Excellent
Item 46	I seek feedback on how people make sense of their care experience	0.83	0.81	Excellent	1	1	Excellent
Item 47	I encourage people to discuss what is important to them	0.83	0.81	Excellent	1	1	Excellent
	**Shared decision-making**						
Item 48	I include the family in care decisions where appropriate and/or in line with the person’s wishes	0.83	0.81	Excellent	1	1	Excellent
Item 49	I work with the person to set health goals for their future	1	1	Excellent	1	1	Excellent
Item 50	I enable people receiving care to seek information about their care from other healthcare professionals	1	1	Excellent	1	1	Excellent
	**Engagement**						
Item 51	I try to understand the person’s perspective	0.83	0.81	Excellent	1	1	Excellent
Item 52	I seek to resolve issues when my goals for the person differ from their perspectives	0.83	0.81	Excellent	1	1	Excellent
Item 53	I engage people in care processes where appropriate	0.67	0.57	Acceptable	1	1	Excellent
	**Having sympathetic presence**						
Item 54	I actively listen to people receiving care to identify unmet needs	0.83	0.81	Excellent	1	1	Excellent
Item 55	I gather additional information to help me support people receiving care	0.83	0.81	Excellent	1	1	Excellent
Item 56	I ensure my full attention is focused on the person when I am with them	1	1	Excellent	1	1	Excellent
	**Providing holistic care**						
Item 57	I strive to gain a sense of the whole person	0.67	0.57	Acceptable	1	1	Excellent
Item 58	I assess the needs of the person, taking account of all aspects of their lives	1	1	Excellent	1	1	Excellent
Item 59	I deliver care that takes account of the whole person	1	1	Excellent	1	1	Excellent
		S-CVI/ave = 0.90			S-CVI/ave = 0.96		

K * = Modified kappa adjusted for chance. K * = (I-CVI − pc)/(1 − pc). I-CVI = observed index adjusted for the content validity index. Pc = probability that the event occurs by chance

**Table 3 healthcare-12-02485-t003:** Items of the questionnaire that needed changes.

Items	Original Terms (English)	Original Terms (Spanish)	Final Changes
Item 18	Support	Favorezco	Apoyo
Item 45	Of their family and carers	De su familia y sus cuidadores	De su familia y el de sus cuidadores
Item 47	I encourage people to discuss what is important to them	Animo a las personas a que compartan lo que es importante para ellas	Animo a que las personas compartan lo que es importante para ellas
Item 52	Person	Persona	Persona cuidada
Item 54	People receiving care	A las personas que reciben cuidados	A las personas que cuido

**Table 4 healthcare-12-02485-t004:** Descriptive statistics of the sample.

Characteristics of the Sample	N = 287	%
**Mean age**	44.5	
**Sex**		
Male	34	11.97
Female	252	87.68
No response	1	0.35
**Professional group**		
Nurse	152	53.52
Physician	75	26.41
Nursing assistant	50	16.90
Manager	3	1.06
Technician	3	1.06
Social worker	3	0.70
Neuropsychologist	1	0.35
**Experience**		
≤10 years	75	26.06
>10 years	212	73.94

**Table 5 healthcare-12-02485-t005:** Descriptive statistics of the construct variables (sum of items for each construct) of the instrument: skewness, kurtosis, ordinal theta, and polychoric correlation coefficient (n = 284).

Construct	Theta	Skewness	Kurtosis	Mean	Std.	1	2	3	4	5	6	7	8	9	10	11	12	13	14	15	16
C1.Professionally competent	0.75	−0.69	0.42	13.32	1.48																
C2. Developed interpersonal skills	0.83	−0.45	−0.69	18.08	1.63	0.55															
C3. Being committed to the job	0.89	−0.45	−0.71	22.26	2.17	0.56	0.67														
C4. Knowing self	0.88	−0.21	−0.78	13.03	1.56	0.48	0.58	0.66													
C5. Clarity of beliefs and values	0.74	−0.04	−0.16	11.30	1.85	0.36	0.49	0.57	0.60												
C6. Skill mix	0.76	0	−0.21	12.15	1.61	0.41	0.50	0.57	0.50	0.65											
C7. Shared decision-making systems	0.82	−0.38	0.19	14.64	3.03	0.30	0.34	0.42	0.38	0.55	0.46										
C8. Effective staff relationships	0.86	−0.54	0.15	11.70	2.21	0.21	0.35	0.43	0.39	0.47	0.44	0.66									
C9. Power sharing	0.88	−0.58	0.02	15.51	2.99	0.23	0.39	0.42	0.41	0.47	0.43	0.70	0.79								
C10. Potential for innovation and risk taking	0.73	0	−0.43	11.76	1.82	0.31	0.47	0.57	0.52	0.52	0.51	0.65	0.64	0.70							
C11. The physical environment	0.86	−0.33	0.13	12.26	1.82	0.40	0.48	0.63	0.60	0.60	0.62	0.30	0.36	0.36	0.52						
C12. Supportive organizational systems	0.88	−0.24	−0.24	16.01	4.07	0.14	0.29	0.38	0.39	0.45	0.40	0.61	0.67	0.73	0.60	0.36					
C13. Working with patients’ beliefs and values	0.88	0	−0.39	16.70	2.10	0.48	0.61	0.75	0.65	0.61	0.61	0.43	0.43	0.37	0.53	0.71	0.40				
C14. Shared decision-making	0.83	−0.37	0.47	12.29	1.73	0.32	0.47	0.62	0.46	0.45	0.45	0.38	0.39	0.37	0.50	0.54	0.35	0.72			
C15. Engagement	0.87	0.1	−0.64	12.82	1.45	0.40	0.55	0.62	0.47	0.43	0.49	0.38	0.36	0.35	0.50	0.49	0.29	0.69	0.74		
C16. Having sympathetic presence	0.87	−0.05	−0.25	12.75	1.52	0.46	0.58	0.67	0.52	0.50	0.54	0.35	0.42	0.37	0.49	0.59	0.36	0.75	0.62	0.65	
C17. Providing holistic care	0.94	−0.49	0.16	13.18	1.59	0.44	0.57	0.68	0.53	0.48	0.51	0.31	0.32	0.31	0.48	0.63	0.30	0.72	0.61	0.68	0.76

Green color: Highest level of reliability. Yellow and orange color: Middle level of reliability. Red color: Lowest level of reliability.

**Table 6 healthcare-12-02485-t006:** Goodness-of-fit indices for various models.

Model	χ^2^	d.f.	Δχ^2^	Δd.f.	Prob. > χ^2^	CFI	SRMR	RMSEA	(RMSEA CL90)
Base model	10,873.50	1711							
Original PCPI-S (Slater 2017)	2716.60	1516	8156.9	195	<0.001	0.87	0.068	0.053	(0.050–0.056)
Spanish PCPI-S (without C10)	2329.18	1364	387.42	152	<0.001	0.89	0.058	0.050	(0.047–0.053)

**Table 7 healthcare-12-02485-t007:** Properties of the revised model of the PCPI-S.

Construct/Item		Variance Extracted Estimate	Reliability	Standardized Loading	t Value
1	**Professionally competent**	0.31	^a^ 0.56		
	I have the necessary skills to negotiate care options		0.20	0.45	8.03
	When I provide care, I pay attention to more than the immediate physical task		0.20	0.45	8.17
	I actively seek opportunities to extend my professional competence		0.53	0.73	15.84
2	**Developed interpersonal skills**	0.41	^a^ 0.73		
	I ensure I hear and acknowledge others’ perspectives		0.61	0.78	23.70
	In my communication I demonstrate respect for others		0.28	0.53	10.84
	I use different communication techniques to find mutually agreed solutions		0.43	0.65	16.01
	I pay attention to how my non-verbal cues impact on my engagement with others		0.31	0.56	11.76
3	**Being committed to the job**	0.45	^a^ 0.80		
	I strive to deliver high-quality care to people		0.39	0.62	15.42
	I seek opportunities to get to know the person and their family in order to provide holistic care		0.44	0.67	17.99
	I go out of my way to spend time with people receiving care		0.52	0.72	21.97
	I strive to deliver high-quality care that is informed by evidence		0.47	0.69	19.45
	I continuously look for opportunities to improve the care experiences		0.43	0.65	17.18
4	**Knowing self**	0.59	^a^ 0.81		
	I take time to explore why I react as I do in certain situations		0.64	0.80	27.75
	I use reflection to check out if my actions are consistent with my ways of being		0.72	0.85	32.99
	I pay attention to how my life experiences influence my practice		0.40	0.63	15.55
5	**Clarity of beliefs and values**	0.41	^a^ 0.68		
	I actively seek feedback from others about my practice		0.34	0.58	12.92
	I challenge colleagues when their practice is inconsistent with our team’s shared values and beliefs		0.41	0.64	15.42
	I support colleagues to develop their practice to reflect the team’s shared values and beliefs		0.50	0.70	18.80
6	**Skill mix**	0.39	^a^ 0.66		
	I recognize when there is a deficit in knowledge and skills in the team and its impact on care delivery		0.43	0.65	15.84
	I am able to make the case when the skill mix falls below acceptable levels		0.41	0.64	15.39
	I value the input from all team members and their contributions to care		0.33	0.58	12.63
7	**Shared decision-making systems**	0.47	^a^ 0.78		
	I actively participate in team meetings to inform my decision-making		0.48	0.69	19.48
	I participate in organization-wide decision-making forums that impact on practice		0.39	0.62	15.24
	I am able to access opportunities to actively participate in influencing decisions in my directorate/division		0.60	0.77	26.10
	My opinion is sought in clinical decision-making forums (e.g., ward rounds, case conferences, discharge planning)		0.43	0.66	17.31
8	**Effective staff relationships**	0.61	^a^ 0.82		
	I work in a team that values my contribution to person-centred care		0.73	0.85	39.02
	I work in a team that encourages everyone’s contribution to person-centred care		0.70	0.84	36.52
	My colleagues positively role model the development of effective relationships		0.39	0.62	15.53
9	**Power sharing**	0.57	^a^ 0.84		
	The contribution of colleagues is recognized and acknowledged		0.49	0.70	20.95
	I actively contribute to the development of shared goals		0.48	0.69	20.53
	The leader facilitates participation		0.58	0.76	27.44
	I am encouraged and supported to lead developments in practice		0.72	0.85	40.88
11	**The physical environment**	0.55	^a^ 0.79		
	I pay attention to the impact of the physical environment on people’s dignity		0.53	0.73	21.78
	I challenge others to consider how different elements of the physical environment impact on person-centredness (e.g., noise, light, heat, etc.)		0.60	0.78	25.93
	I seek out creative ways of improving the physical environment		0.52	0.72	21.10
12	**Supportive organizational systems**	0.54	^a^ 0.85		
	In my team we take time to celebrate our achievements		0.48	0.69	19.90
	My organization recognizes and rewards success		0.55	0.74	24.19
	I am recognized for the contribution that I make to people having a good experience of care		0.46	0.68	18.82
	I am supported to express concerns about an aspect of care		0.67	0.82	33.83
	I have the opportunity to discuss my practice and professional development on a regular basis		0.55	0.74	23.96
13	**Working with patients’ beliefs and values**	0.53	^a^ 0.82		
	I integrate my knowledge of the person into care delivery		0.52	0.72	23.49
	I work with the person within the context of their family and carers		0.59	0.77	28.21
	I seek feedback on how people make sense of their care experience		0.46	0.67	19.59
	I encourage people to discuss what is important to them		0.56	0.75	26.54
14	**Shared decision-making**	0.53	^a^ 0.77		
	I include the family in care decisions where appropriate and/or in line with the person’s wishes		0.55	0.74	22.96
	I work with the person to set health goals for their future		0.55	0.74	23.00
	I enable people receiving care to seek information about their care from other healthcare professionals		0.47	0.69	18.91
15	**Engagement**	0.55	^a^ 0.79		
	I try to understand the person’s perspective		0.56	0.75	24.05
	I seek to resolve issues when my goals for the person differ from their perspectives		0.51	0.71	21.07
	I engage people in care processes where appropriate		0.60	0.77	26.36
16	**Having sympathetic presence**	0.58	^a^ 0.80		
	I actively listen to people receiving care to identify unmet needs		0.68	0.83	34.05
	I gather additional information to help me support people receiving care		0.49	0.70	20.46
	I ensure my full attention is focused on the person when I am with them		0.56	0.75	24.54
17	**Providing holistic care**	0.69	^a^ 0.87		
	I strive to gain a sense of the whole person		0.62	0.79	30.09
	I assess the needs of the person, taking account of all aspects of their lives		0.67	0.82	35.45
	I deliver care that takes account of the whole person		0.78	0.88	48.93

^a^ Composite reliability.

## Data Availability

The data presented in this study are available on request from the corresponding author.

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
