# Peer review of "Translation, Cultural Adaptation, and Validation of the Spanish Version of the Person-Centred Practice Inventory-Staff (PCPI-S)"

_healthcare, 2024, doi:10.3390/healthcare12232485_

Round 1
Reviewer 1 Report
Comments and Suggestions for Authors
Congratulations on a well-writen piece of research. I only have one comment: When you state that the word "care" has been kept in the Spanish version, it can suggest that you are keeping it in English. Include the Spanish term in brackets to avoid misunderstanding.
Author Response
Please, see the attachment.
Thank you very much.

Reviewer 2 Report
Comments and Suggestions for Authors
This manuscript focuses on the translation, cultural adaptation, and psychometric validation of the Spanish version of the Person-Centred Practice Inventory-Staff (PCPI-S). Overall, the manuscript is well-structured and the research design is sound, with two main phases: translation and cultural adaptation, and psychometric assessment. The author rigorously follows recognized cultural adaptation guidelines to present the process of converting the tool from English to Spanish. The manuscript provides a detailed analysis of the tool's structure, internal consistency, validity, and test-retest reliability, demonstrating that the Spanish version of PCPI-S is a valid instrument for assessing staff perceptions of person-centred practices in the Spanish healthcare context. However, there are several areas in which the manuscript could be further strengthened.
First, in the introduction, the manuscript lacks a clear emphasis on the significance and value of the research. The introduction should better highlight the specific benefits that the Spanish version of the PCPI-S could bring to the local healthcare system. For example, it would be useful to explain how the tool can improve the assessment of person-centred practices in Spanish-speaking healthcare settings, and how it can ultimately contribute to enhancing patient care or healthcare worker practices. Without this contextual explanation, readers may question whether the translation and validation of the instrument is truly necessary, especially if other similar tools are already in use or if the need for such a tool in Spanish-speaking countries has not been clearly established.
Second, I cannot find Table 2 in this manuscript. By the way, some items (0.2-0.4) have low retest reliability, and the relevant possible reasons should be clarified in the discussion.
Third, to enhance the clarity and depth of the manuscript, I recommend that the authors provide an in-depth discussion on the high correlation between "Clarity of beliefs and values" and "Skill-Mix." They should explain whether the correlation is expected based on theoretical grounds or if it points to potential issues with conceptual overlap. Additionally, they should address how this correlation affects the validity of the tool and whether any adjustments are needed in future iterations of the tool or in its interpretation.
Finally, I suggest that the authors expand the discussion on the applicability of the Spanish version of PCPI-S to multiple Spanish-speaking countries. They should acknowledge that the tool's validation in a single country is a starting point, and that further research is required to determine whether it holds the same psychometric properties across different Spanish-speaking regions. Proposing multi-country validation studies or cross-cultural research in diverse healthcare settings would strengthen the manuscript’s claims about the tool's broader applicability.
Reviewer 3 Report
Comments and Suggestions for Authors
Thanks to the authors for sharing their manuscript. Appreciating the conducted research, I would like to make a few comments:
Given the number of items and the sample of the study, it would be justified to provide an analysis of statistical power.
Although the authors conclude that the factor structure of the PCPI-S is adequate, the CFI still cannot be called acceptable even by the mildest criteria. Moreover, the authors themselves rely on strict criteria highlighted by Hu and Bentler (1999). In this regard, questions arise about the results of factor analysis. Can they be considered reliable? Is the study sample too small? Or is it worth evaluating modified tool models?
There are very few sources in the list of references for the last two or three years. This raises questions about the relevance and scientific novelty of the research. I suggest that the authors study the literature more thoroughly and refine the introduction and discussion based on more modern and relevant sources.
I hope that my comments will help improve the manuscript.
Sincerely yours, reviewer.
Round 2
Reviewer 2 Report
Comments and Suggestions for Authors
I have no new comments.
Reviewer 3 Report
Comments and Suggestions for Authors
Dear authors, thank you for sharing the revised manuscript. I believe that my comments have been taken into account and recommend the manuscript for publication.